# Rehardening and the Protective Effect of Gamma-Polyglutamic Acid/Nano-Hydroxyapatite Paste on Surface-Etched Enamel

**DOI:** 10.3390/polym13234268

**Published:** 2021-12-06

**Authors:** Nai-Chia Teng, Aditi Pandey, Wei-Hsin Hsu, Ching-Shuan Huang, Wei-Fang Lee, Tzu-Hsin Lee, Thomas Chung-Kuang Yang, Tzu-Sen Yang, Jen-Chang Yang

**Affiliations:** 1School of Dentistry, College of Oral Medicine, Taipei Medical University, Taipei 11031, Taiwan; tengnaichia@hotmail.com; 2Department of Dentistry, Taipei Medical University Hospital, Taipei 11031, Taiwan; jollyhuangtw12@gmail.com; 3Graduate Institute of Nanomedicine and Medical Engineering, College of Biomedical Engineering, Taipei Medical University, Taipei 11052, Taiwan; aditi8293@tmu.edu.tw (A.P.); tzu6415@hotmail.com (T.-H.L.); 4Department of Chemical Engineering and Biotechnology, National Taipei University of Technology, Taipei 106, Taiwan; haha780513@gmail.com (W.-H.H.); ckyang@ntut.edu.tw (T.C.-K.Y.); 5School of Dental Technology, College of Oral Medicine, Taipei Medical University, Taipei 110, Taiwan; weiwei@tmu.edu.tw; 6Graduate Institute of Biomedical Optomechatronics, Taipei Medical University, Taipei 11031, Taiwan; tsyang@tmu.edu.tw; 7International Ph.D. Program in Biomedical Engineering, College of Biomedical Engineering, Taipei Medical University, Taipei 11031, Taiwan; 8Research Center of Biomedical Device, Taipei Medical University, Taipei 11052, Taiwan; 9Research Center of Digital Oral Science and Technology, Taipei Medical University, Taipei 11052, Taiwan

**Keywords:** dental erosion, nano-hydroxyapatite, γ-poly glutamic acid, atomic force microscope, surface microhardness recovery

## Abstract

Many revolutionary approaches are on the way pertaining to the high occurrence of tooth decay, which is an enduring challenge in the field of preventive dentistry. However, an ideal dental care material has yet to be fully developed. With this aim, this research reports a dramatic enhancement in the rehardening potential of surface-etched enamels through a plausible synergistic effect of the novel combination of γ-polyglutamic acid (γ-PGA) and nano-hydroxyapatite (nano-HAp) paste, within the limitations of the study. The percentage of recovery of the surface microhardness (SMHR%) and the surface parameters for 9 wt% γ-PGA/nano-HAp paste on acid-etched enamel were investigated with a Vickers microhardness tester and an atomic force microscope, respectively. This in vitro study demonstrates that γ-PGA/nano-HAp treatment could increase the SMHR% of etched enamel to 39.59 ± 6.69% in 30 min. To test the hypothesis of the rehardening mechanism and the preventive effect of the γ-PGA/nano-HAp paste, the surface parameters of mean peak spacing (Rsm) and mean arithmetic surface roughness (Ra) were both measured and compared to the specimens subjected to demineralization and/or remineralization. After the treatment of γ-PGA/nano-HAp on the etched surface, the reduction in Rsm from 999 ± 120 nm to 700 ± 80 nm suggests the possible mechanism of void-filling within a short treatment time of 10 min. Furthermore, ΔRa-I, the roughness change due to etching before remineralization, was 23.15 ± 3.23 nm, while ΔRa-II, the roughness change after remineralization, was 11.99 ± 3.90 nm. This statistically significant reduction in roughness change (*p* < 0.05) implies a protective effect against the demineralization process. The as-developed novel γ-PGA/nano-HAp paste possesses a high efficacy towards tooth microhardness rehardening, and a protective effect against acid etching.

## 1. Introduction

Dental enamel, the hardest tissue of the human body, comprises nanoscale hydroxyapatite (HAp) forming the major mineral portion [1,2], with a parallel arrangement of enamel prisms [3,4,5]. Tooth decay, including dental caries and dental erosion caused by demineralization in an acidic environment, can lead to the destruction of the nanostructure of enamel [6,7]. Dental caries are a complex phenomenon associated with the demineralization of HAp from hard tissues by bacteria-metabolized acids [8]. Caries-associated erosion, a dynamic multifactorial disease, begins with a white spot lesion or non-cavitated enamel lesion, which then becomes soft, rough, and cavitated when reaching the dentin–enamel junction, leading to lateral progression [9]. On the other hand, dental erosion is a chemical dissolution process of HAp crystals owing to the action of acids of intrinsic (gastric reflux or disorders) or extrinsic (food or drinks) origin, [10,11,12,13] but not involving bacterial plaque acid or that which is directly associated with mechanical or traumatic factors [14]. The erosion, an irreversible loss of hard dental tissue, is manifested initially as dental sensitivity, proceeding towards enamel softening, dissolution, and eventually tooth structure loss [15]. With the increased popularity of acidic diets, including drinks and food, dental erosion has a widespread and severe impact. Therefore, in order to assist the tooth repair, the reversibility of progressive deterioration could be mediated by a remineralization process under external sources of calcium and phosphate ions, promoting ion deposition into the decayed enamel crystal cavity. This approach, when carried out in clinically relevant conditions, might result in a significant contribution towards dental research [15,16,17].

It is noteworthy that casein phosphopeptides-amorphous calcium phosphate (CPP-ACP) is a phosphopeptide-containing complex of calcium and phosphate ions, and therefore a source of bioavailable Ca^2+^ aiding in the remineralization of hard tissues [18,19]. CPP not only stabilizes the amorphous nature of ACP, preventing the phase transformation into a low solubility of crystalline HAp [18,20,21,22], but also extends the remineralization time period of ACP owing to its good mucoadhesion [23]. CPP-ACP is known for its successful incorporation into commercial dental care products such as sugar-free chewing gums [24] and tooth mousse to reduce enamel erosion through its anticariogenicity in the remineralization of enamel lesions [12,25]. However, the protein casein occasionally has restricted use among people sensitive and/or allergic to milk [26]. Gamma-poly (glutamic acid) (γ-PGA) is a natural anionic peptide produced most notably by the fermentation of *Bacillus subtilis* [27]. γ-PGA is a hydrophilic, biodegradable polymer, which has applications in tissue engineering and drug delivery [28,29,30]. Various studies reported the use of γ-PGA as a nutrition supplement in dietary products for enhanced Ca^2+^ solubility, explaining its role in increased calcium absorption in the small intestine [31,32]. Another widely available material, nano-HAp, has applications in biomedical areas such as dentistry and orthopedics, [33] owing to its excellent biocompatible, osteoconductive, and bioactive behavior [34,35]. The enamel has a complex structure for remodeling and comprises basic building blocks of ~20–40 nm HAp nanoparticles. Thus, for enamel remineralization by minerals, the use of synthetic HAp may also be beneficial as a restorative material [36] with enhanced mechanical properties [37,38]. Nano-HAp is used in applications such as the repair of initial caries-like lesions [39,40]. Compared with ACP-based materials, nano-HAp has the advantage of higher remineralization of initial enamel lesions [41].

Considering the aforementioned information, a novel approach for enhancing the rehardening potential of nano-HAp by incorporating a negatively charged complexing agent of γ-PGA was reported in this study. Furthermore, the protective effect against the etching of teeth by the application of γ-PGA/nano-HAp paste was also investigated in this study.

## 2. Materials and Methods

### 2.1. Preparation of Tooth-Repairing Paste

The commercial nano-hydroxyapatite (nano-HAp, Ca_10_(PO_4_)_6_(OH)_2_ with average particle size less than 200 nm) was purchased from Sigma-Aldrich, Burlington, MA, USA. Gamma-poly (glutamic acid) (γ-PGA, H from, MW 1~10 kDa) was obtained from Vedan, Taipei, Taiwan. Acetic acid (CH_3_COOH, MW 60.05) was purchased from Shimakyu’s Pure Chemicals, Osaka, Japan. To prepare the tooth-repairing paste, 0.27 g γ-*PGA* was first dissolved in 5 mL DI water, then 0.50 g nano-HAp was added to the solution and stirred overnight.

### 2.2. Etched Enamel Preparation

Extracted human teeth were collected after informed consent had been obtained under a protocol permitted from the Joint Institutional Review Board, N201805095, Taipei Medical University. The teeth were cleaned thoroughly and stored in normal saline solution (Taiwan Otsuka Pharmaceutical Co., Taipei, Taiwan) at 4 °C. Enamel discs were cut from the buccal side of the crown by means of a low-speed 0.15 mm diamond saw (Three A Co., Ltd., Taoyuan City, Taiwan). Each enamel disc was embedded into an acrylic resin (Ortho-Jet; Lang Dental Manufacturing Co., Wheeling, IL, USA), then the buccal side of each enamel disc was sanded with 1000-grit and 1200-grit water sandpaper at 500 rpm to create a smooth surface. This was followed by the application of nail polish on the enamel surface and the designed demineralization and remineralization, as described further.

The experimental group was 9 wt% γ-PGA/nano-HAp paste, and control group was GC Tooth Mousse^®^ (Recaldent™ GC Corporation, Tokyo, Japan). To evaluate the effectiveness of aforesaid repairing pastes, the surface of tooth enamel was first etched by 1M acetic acid for 3 min, applied to each specimen on the pre-etched surface for various time periods, and then Vickers hardness number (VHN) was measured. Figure 1 shows the detailed procedure of tooth etching and treatment. The refurbishing pastes were applied on the surface of teeth samples for treatment for 10 min each, thrice up to 30 min, with 2 min-long intervals.

### 2.3. Surface Microhardness Recovery (SMHR) Measurement

The Vickers hardness (HV) test of each specimen was evaluated using a Mitutoyo microhardness tester MVK G1 (Mitutoyo Corp., Tokyo, Japan), with a square-based pyramid-shaped diamond indenter under a full load of 100 g for 15 *s* at room temperature. Six indentations were made on the polished surface of each treated specimen (N = 6), at separate locations no closer than 1 mm to adjacent indentations or the specimen periphery, according to the guidelines of ASTM C1327-08 standard test methods for Vickers indentation hardness of advanced ceramics [42]. The diagonal of the resulting indention was measured under the microscope and the HV value (VHN) was displayed on the digital readout of the lens. The rehardening potential was measured and calculated to obtain the percentage of surface microhardness recovery (SMHR%) from etched teeth according to Equation (1).
(1)$$\mathrm{SMHR}\%=\frac{{\mathrm{VHN}}_{\mathrm{after}\;\mathrm{remineralization}}-{\mathrm{VHN}}_{\mathrm{after}\;\mathrm{acid}\;\mathrm{erosion}}}{{\mathrm{VHN}}_{\mathrm{original}\;\mathrm{tooth}}-{\mathrm{VHN}}_{\mathrm{after}\;\mathrm{acid}\;\mathrm{erosion}}}*100$$

### 2.4. Surface Profile Measurement by Atomic Force Microscopy

Atomic force microscopy (AFM) was used to analyze the surface topography of enamel. The characterization was performed using a PSIA XE-100 (Park Systems Corp., Suwon, Korea) AFM system at ambient temperature and humidity. Aluminum-coated silicon cantilever was used in a non-contact mode to probe the surface across an area of 5 × 5 µm on the sound, acid-etched, and re-mineralized samples. The scanning rate of 0.25 Hz was used. Surface analysis was performed to obtain the mean peak spacing (Rsm) and mean arithmetic surface roughness (Ra) using the software of the AFM instrument.

### 2.5. Statistical Analysis

Group means and standard deviations (SD) were calculated. Data were compared using the GraphPad Prism software by the one-way analysis of variance (ANOVA) and post hoc Tukey’s test, and *p* < 0.05 indicated statistical significance.

## 3. Results and Discussions

### 3.1. Percent Surface Microhardness Recovery (SMHR%) Analysis

The microhardness of the sound, etched, and remineralized enamel surfaces are depicted in Table 1. The microhardness of the sound sample was found to be 361.16 ± 9.21 VHN, which reduced to 218.41 ± 6.87 VHN after acid-etching. This was followed by a substantial increment in microhardness after treatment with 9 wt% γ-PGA/nano-HAp paste. The SMHR% were found to be 30.10 ± 5.53%, 36.27 ± 6.95%, and 39.59 ± 6.69% after 10, 20, and 30 min of treatment, respectively, and were found to be higher (which is statistically significant) than that of the GC Tooth Mousse^®^-treated samples, portraying the rehardening potential of the refurbishing paste.

Tooth erosion has gained attention for contributing to enamel wear [11]. Owing to the limited mineral ions in the oral remineralizing solutions (for building up the enamel), a considerably large number of inorganic ions need to be externally supplied for the enhanced precipitation of mineral content into the demineralized enamel pores/lesions, acid resistance, and mechanical protection against erosion [26,43]. Hence, this could be beneficial when encountering demineralization-imposed challenges (irreversible destructions). Based on this concept, many tooth-repairing products have been developed. In this study, we utilized γ-PGA/nano-HAp as the material for investigating tooth-erosion studies. When compared to CPP (having uncharged groups), γ-PGA, containing repeat units of charged glutamate (-Glu-Glu-Glu-) linked in between the γ-carboxylic acid and α-amino functional groups, possesses an affinity for enamel (because of its phosphate and calcium minerals) [44]. It was also reported that nano-HAp is effective in the biomimetic repair of enamel exposed to acid attacks [38]. Nano-HAp was reported to deposit strongly on the etched surface of enamel, inhibiting the subsequent mineral loss [45]. However, it was also suggested that nano-HAp can remineralize the outer layer of an enamel caries lesion, but without complete remineralization. Therefore, combining it with another material may enhance its remineralization potential.

The typical in vitro demineralization–remineralization studies for simulating the cariogenic challenges in the mouth are classified into biofilm and chemical models [46]. Furthermore, chemical models are categorized into pH-cycling and simple mineralization models for artificial dental caries and dental erosion, respectively [47]. We conducted the simple mineralization model using an acid-eroded enamel surface to assay each repairing paste for a total duration of 30 min (i.e., each treatment of 10 min repeated three times). There was no significant difference in the SMHR% among the groups with increasing rehardening time (from 10 to 30 min), though. However, an SMHR of ~2-folds higher was obtained for the 9 wt% γ-PGA/nano-HAp treatment group (39.59 ± 6.69% SMHR), compared to those treated with the commercially available GC Tooth Mousse^®^ (12.32 ± 0.16% SMHR). Henceforth, detailed clinical studies were required for in-depth analysis and confirmation.

The active ingredient of GC Tooth Mousse^®^ is CPP-ACP, which, when placed on the tooth surface, interacts with various ions, diffusing into the enamel and thereby producing sub-surface mineral gains [48]. F. Hua et al. reported a novel carrier-based delivery system of PAA-ACP@aMSN based on amine-functionalized pore mesoporous silica (aMSN) loaded with polyacrylic acid (PAA) for amorphous calcium phosphate (ACP) [49]. The surface microhardness recovery ratio showed no significant difference between CPP-ACP and PAA-ACP@aMSN after a 7-day remineralization procedure on the enamel white spot lesions. Moreover, it was reported that nano-HAp was found to be more effective than that of CPP-ACP paste at enhancing the mineral content of enamel and increasing its microhardness [50,51,52]. S. Vyavhare et al. reported a similar trend for comparing different remineralizing agents on surface microhardness recovery. The reported SMHR% for nano-HAp (10%), CPP-ACP (10%), NaF (1000 ppm), and deionized water (negative control) after 6 days of pH cycling were 37.06 ± 7.32%, 18.17 ± 7.70%, 37.68 ± 7.06%, and 16.52 ± 8.77%, respectively [53]. Furthermore, γ-PGA/tricalcium phosphate was recently reported to be more effective than CPP-ACP at enhancing the microhardness of bovine dentin samples after 2 weeks of pH cycling remineralization [53]. S. Huang et al. reported 35.4% of SMHR in 144 min of treatment time in enamel remineralization under the pH-cycling model for dental caries, using nano-HAp combined with Galla chinensis (GCE) [54]. A synergistic effect of nano-HAp/GCE on the treatment group showed more mineral deposition in the lesion body and less lesion depth. Unlike the long treatment time (ranging from 144 min to 2 weeks) previously reported from remineralizing agents in the pH-cycling model, we reported the high SMHR% after a short treatment period of 10–30 min in a simple demineralization model. In fact, it is widely accepted that the biomineralization pathway in living organisms is governed by a heterogeneous nucleation model of ACP-mediated HAp crystallization rather than direct HAp crystallization [54]. S. Jiang et al. claimed that the crystallization process comprises three stages: (1) induction period; (2) crystallization stage; (3) post-crystallization stage. The mineral phases of precipitation at stage I (at 2 h and 4 h, ACP), II (at 5.5 h and 7 h, weakly crystallized HAp), and III (at 10 h, crystallized HAp), were each characterized ex situ by FTIR spectra and XRD patterns [55]. In contrast to the long induction period of 2 h for ACP formation in the supersaturation’s mineral phase, with 10–30 min of treatment time of γ-PGA/nano-HAp repairing paste for a high SMHR%, it can be concluded that there may exist a probable synergy between γ-PGA and nano-HAp for the Ca^+^ and PO_4_^−^ ions’ deposition and the consecutive rehardening. Furthermore, we hypothesize the possible occurrence of an additional mechanism for the rehardening of etched surfaces besides the typical remineralization with calcium and phosphate ion deposition.

### 3.2. Effect of γ-PGA/Nano-HAp Paste on AFM Surface Parameters

The AFM images were obtained for analyzing the topography and structural information. The assessment of the linear spacing parameter (Rsm values) of the surfaces of acid-etched tooth samples before and after remineralization are represented in the form of 3D plots in Figure 2. The Rsm values were decreased from 990 ± 120 nm to 700 ± 80 nm before and after remineralization, respectively.

Figure 3 depicts the surface roughness (Ra) for the polished (sound), acid-etched (E), and remineralized (R) enamels at varying time points. It can be illustrated from the micrographs in Figure 3a that the sound enamel roughness (8.28 ± 8.84 nm) increases upon inducing demineralization by process of acid-etching as E-5 (20.30 ± 0.71) and E-10 (31.43 ± 2.39 nm), at each stage for 5 min. Post-remineralization, the roughness after the application of γ-PGA/nano-HAp paste for 20 min, shown as R-20, was obtained as 31.81 ± 1.33 nm (no significant change when compared with E-10, *p* > 0.05). Furthermore, upon etching (5 min each) of the remineralized samples, the roughness values of 33.62 ± 1.43 nm as Er-5 and 43.84 ± 2.57 nm as Er-10 were obtained.

The roughness change was assessed before and after the first acid etching (E-10 min) of the sound samples as 23.15 ± 3.23 nm (ΔRa-I = E-10 min − Sound) (Figure 3b). Later, upon second etching, the difference in the roughness of the second acid etching (ΔRa-II = Er-10 min − R-20 min) between the etched-remineralized (Er-10 min) and remineralized (R-20 min) samples was found to have reduced (*p* < 0.05) to 11.99 ± 3.90 nm, hence indicating a preventive action of the repairing paste.

AFM has been widely used to investigate the erosion of enamel and dentin [56]. Rsm designates the mean width of the profile elements for roughness, indicating the average length of the profile element along the sampling length. A typical nano-HAp crystallite has a needle-like morphology [57]. Therefore, our hypothesis for achieving a high SMHR% in a short treatment time is by filling the cavity of the etched enamel surface with nano-HAp crystallites in a horizontal settling/sitting way. Figure 4 portrays the plausible graphical representation of the proposed fast-repairing model using roughness profile analysis. The reduction in Rsm in the AFM profile can perhaps be attributed to the void-filling of the nano-HAp particles by shortening the original length of the profile element. Therefore, it is assumed that the void-filling effect of γ-PGA/nano-HAp might plays a major role in the early stage of the rehardening process.

The roughness value in terms of mean arithmetic surface roughness (Ra) was evaluated for the enamel surface after its de- and remineralization, in order to evaluate the preventive action of γ-PGA/nano-HAp repairing paste against erosion. There was a significant reduction between the first and second roughness change (ΔRa-I > ΔRa-II). AFM images of the enamel surface treated with repairing paste resulted in fewer morphological changes in the second stage of etching, implying the protective effect of γ-PGA/nano-HAp against acid demineralization. On the other hand, the surface roughness before and after applying the repairing paste displayed no significant difference within the limitations of treatment as short as 20 min.

M. Ceci et al. utilized the AFM technique to investigate the prevention and remineralization effects of CPP-ACP on enamel erosion under soft drink exposure [58]. The efficacy of CPP-ACP was attributed to its buffering of the activity of free calcium and phosphate ions due to the retention of CPP-ACP clusters on the tooth surface, thereby helping to maintain a supersaturation to offset the demineralization and enhance the remineralization. The surface roughness of the samples was found to have changed little, irrespective of the use of the protective/repairing agent (CPP-ACP) before or after soft drink exposure, or even between two erosion cycles [58]. Similar findings about the insensitive surface roughness change before and after remineralization by CPP-ACP were also observed in the case of applying the γ-PGA/nano-HAp paste within the short treatment time. The long induction time for ACP formation in the supersaturation’s mineral phase could be one of the possible reasons for this result. Amorphous calcium phosphates (ACP) have been reported to possess buffering capacity. Therefore, in this study, the HAp was believed to release Ca^+^, H_2_PO_4_^−^ and HPO_4_^2−^ ions under acidic conditions, thereby helping to achieve a possible preventive effect against acid erosion [59]. Furthermore, the plausible mechanism of the γ-PGA/nano-HAp binding to the enamel surface may be the anionic side chain residues (COO^−^ groups of γ-PGA), which bind with the cationic entity (such as positively charged Ca^+^ ions) of the enamel. Moreover, it has been recently reported that γ-PGA has a protective role in forming a layer of coating on the tooth surface to protect it from demineralization and promote remineralization [44]. A synergistic effect of γ-PGA and nano-HAp on enamel-repairing efficacy and preventive action might be possible and beneficial in cases of both dental erosion and caries, but more detailed investigations are required.

## 4. Conclusions

Laying emphasis on tooth erosion, various active ingredients have made significant advanced contributions during the last few decades. Along the same line, in this study, the effect of a novel combination of γ-PGA/nano-HAp surface-etched enamels was analyzed. Upon applying γ-PGA/nano-HAp paste to surface-etched enamel, the higher SMHR% relative to the commercial product GC Tooth Mousse^®^’s application on the etched samples justifies the role of γ-PGA/nano-HAp in rehardening potential. Furthermore, a void-filling mechanism of the γ-PGA/nano-HAp paste was also concluded from the ~41.42% reduced Rsm values as a plausible reason for its decrease before and after the remineralization process. The significant reduction in roughness difference ΔRa before and after remineralization explains the protection against erosion due to the action of γ-PGA/nano-HAp application on the enamel surface. Thus, the as-formed γ-PGA/nano-HAp paste enhanced the tooth’s microhardness and acid-resistant efficiency. However, further studies are required for a clinical outcome of this work.

## Figures and Tables

**Figure 1 polymers-13-04268-f001:**
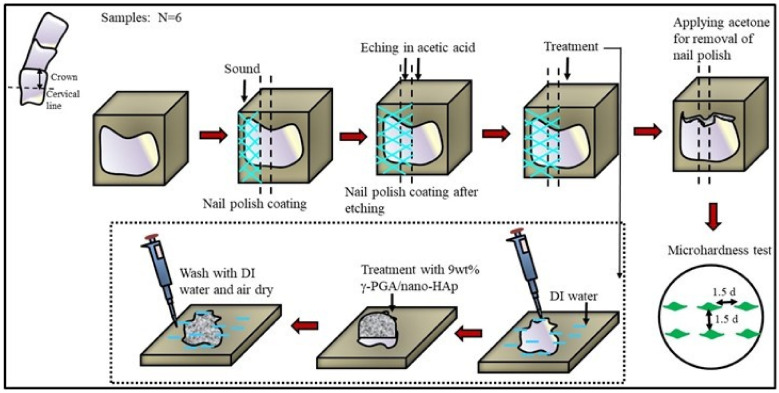
Scheme for demineralization and remineralization of tooth etching, treatment, and microhardness measurement.

**Figure 2 polymers-13-04268-f002:**
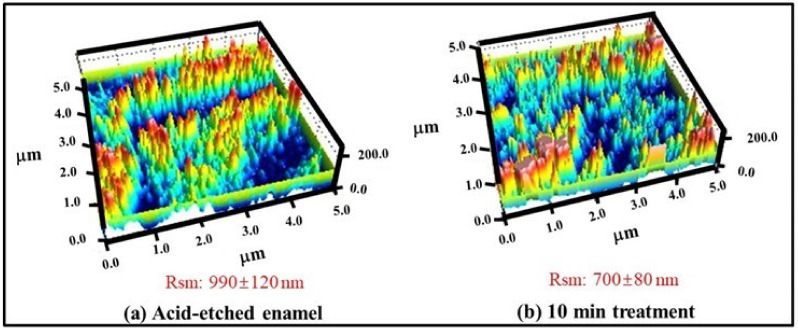
The representative 3-D plots of surface topology and spacing parameters (Rsm) for (**a**) acid-etched enamel surface and (**b**) after 10 min remineralization treatment.

**Figure 3 polymers-13-04268-f003:**
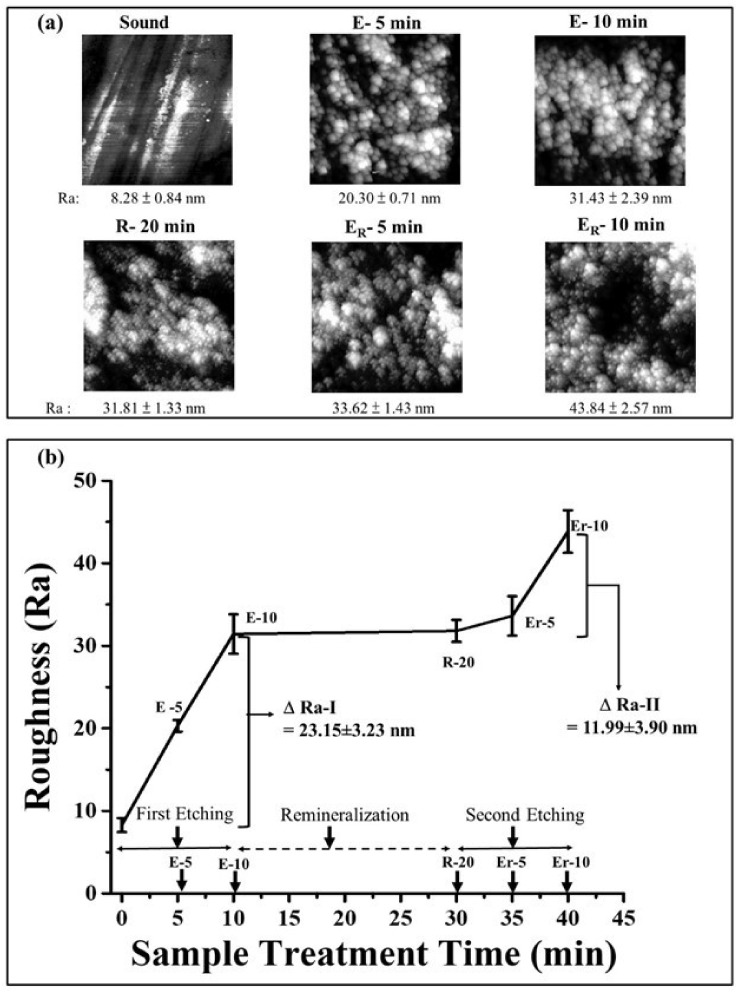
(**a**) AFM micrographs and (**b**) graphical representation of roughness of tooth enamel after demineralization and remineralization.

**Figure 4 polymers-13-04268-f004:**
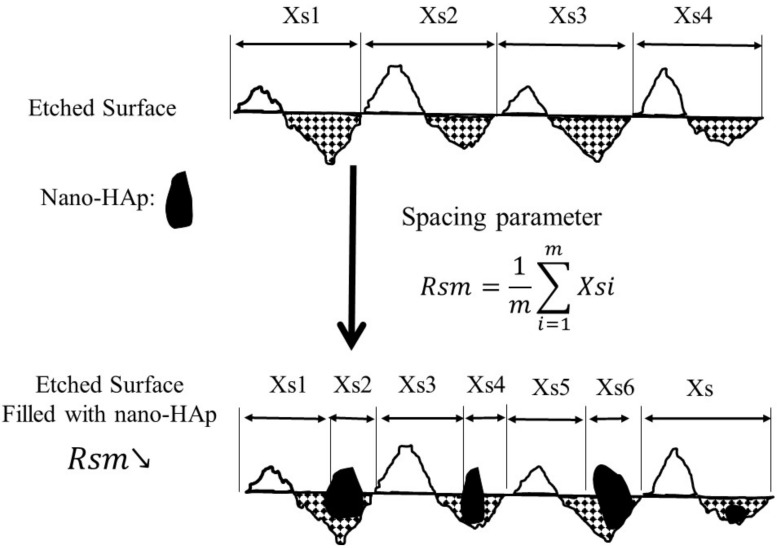
Proposed fast-repairing model for the AFM spacing parameter.

**Table 1 polymers-13-04268-t001:** VHN and SMHR% of γ-PGA/nano-HAp and GC Tooth Mousse^®^.

Group	9 wt%-γPGA/Nano-HAp		GC Tooth Mousse	
	VHN	SMHR%	VHN	SMHR%
**Sound enamel**	361.16 ± 9.21 ^a^		332.95 ± 2.05 ^a^	
**Etched enamel**	218.41 ± 6.87 ^b^		208.29 ± 2.01 ^b^	
**Treat 10 min**	261.31 ± 0.32 ^c^	30.10 ± 5.53	221.77 ± 0.62 ^c^	10.81 ± 1.13
**20 min**	270.11 ± 2.21 ^c^	36.27 ± 6.95	223.15 ± 1.81 ^c^	11.92 ± 0.17
**30 min**	274.84 ± 1.75 ^c^	39.59 ± 6.69	223.65 ± 2.21 ^c^	12.32 ± 0.16

Note: Values were shown as the mean±standard deviation. Only the mean values followed by the different superscript letters significantly differ (*p* < 0.05) according to post hoc test.

## Data Availability

MS thesis of Wei-Hsin Hsu.

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
