# Peer review of "Rehardening and the Protective Effect of Gamma-Polyglutamic Acid/Nano-Hydroxyapatite Paste on Surface-Etched Enamel"

_polymers, 2021, doi:10.3390/polym13234268_

Round 1

Reviewer 1 Report

Dear Author 

The Paper and experiment are quite well. Just want to share a few pieces of literature for strengthening the introduction. 

1) Amin, Faiza, et al. "Effect of Nanostructures on the Properties of Glass Ionomer Dental Restoratives/Cements: A Comprehensive Narrative Review." Materials 14.21 (2021): 6260.

2) Chen, Lijie, et al. "Hydroxyapatite in Oral Care Products—A Review." Materials 14.17 (2021): 4865.

3) Anthoney, Daud, et al. "Effectiveness of thymoquinone and fluoridated BioACTIVE glass/nano-oxide contained dentifrices on abrasion and dentine tubules occlusion: an ex vivo study." European journal of dentistry 14.01 (2020): 045-054.

All the above references information is quite good for the introduction. 

Figure 2, 3 and 4 resolution is not good. improve them. 

Before concluding this study author have to share what is the limitation of this study? 

In conclusion: Authors have to give good outcomes of the study. What is the clinical outcomes of this work?

Reviewer 2 Report

The manuscript can be accepted after minor revisions for the following points.

Avoid writing any abbreviations in keywords.

The experimental part needs to be rewritten with more clarifications.

SEM should be performed for 279
enamel after de-mineralization and re-mineralization.

Reviewer 3 Report

Dear author

This is good paper but should be improved it. Please see below my sugestions from below:

Please prepare better aim

“The nano-hydroxyapatite (nano-HAp, Ca10(PO4)6(OH)2 with particles size less than 200nm

 nm)” sorry but I am not sure nano-hydroxyapatite please see again nosize definition

Figure 1: should be schema of…….

Statistical analysis missing software

Round 2

Reviewer 1 Report

Dear Authors

Well revised. 

Reviewer 3 Report

Q1- Please prepare better aim and mention type of tests made

Q2-“The nano-hydroxyapatite (nano-HAp, Ca10(PO4)6(OH)2 with particles size less than 200nm)” sorry but I am not sure nano-hydroxyapatite please see again nano size definition

Please change title or please include test to measure this powder I mean the granulometry by laser, BET analysis for surface areea or other. It is not nano even you spent money for this. Please ask your colleagues.

This manuscript is a resubmission of an earlier submission. The following is a list of the peer review reports and author responses from that submission.

Round 1

Reviewer 1 Report

TOO MANY MARKS in the authors list, please remove it, the template for MDPI articles is very clear in this aspect.

The  title of the article  “Effect of gamma-polyglutamic acid/nano-hydroxyapatite paste on rehardening and prevention of surface etched enamel” is unclear. Dentistry and especially preventive dentistry is trying to find an ideal material for dental care, this study comes to the rescue with new information on this topic. The reversibility of progressive deterioration could be mediated by the remineralization process under external sources of calcium and phosphate ions to promote ion deposition in the damaged crystalline cavity of the enamel. The introduction describes the process of dental caries as well as the process of dental erosion. The advantages, indications and contraindications of case in phosphopeptides - amorphous calcium phosphate (CPP-ACP) are described. It is proving to be effective for remineralizing hard tissues. It also presents nano-HAp which is widely used in several medical fields, including dentistry, being a biocompatible, osteoconductive and bioactive material. The materials (nano-hydroxyapatite (nano-HAp, Ca10 (PO4)6(OH)2 with particles size less than 200 nm), Gamma-poly(glutamic acid),  Acetic acid (CH3COOH, MW 60.05) ) used were presented in the materials and methods section, the place of purchase being mentioned.

 In the section of materials and methods were described: toothpaste preparation, etched enamel preparation, surface microhardness recovery (SMHR) measurement, surface profile measurement by atomic force microscopy, statistical analysis. The study was performed on extracted teeth. The methods of cleaning and storing teeth were clearly described, as well as the methods for performing demineralization and remineralization of teeth. The engraving and dental treatment procedure was also presented in the form of an image.  The microhardness tester used to evaluate the hardness of each sample was described. The reinforcement potential was measured and calculated. Atomic force microscopy was also used to analyze the topography of the enamel surface. The system by which the characterization was made was also described. The results were compared using unidirectional variation analysis and the Turkey post-hoc test. Percentage analysis of surface microhardness recovery ( SMHR%) was described in the results and discussions section and presented using a table. The assessment of the linear spacing parameter (Rsm values) of the surfaces of acid-etched  tooth samples before and after remineralization are represented in the form of 3D plots. The surface roughness for polished, acid-etched and remineralized enamels at different times, as well as the change in roughness that was assessed before and after the first acid etching are presented with easy-to-understand figures. The proposed rapid repair model using roughness profile analysis was presented using an easy-to-understand figure.

 This study addressed a new technique for creating nano-HAp reinforcement potential by incorporating a negatively charged Æ´-PGA complexing agent. In addition, this study also investigated the effect of prevention against tooth etching by applying Æ´-PGA / nano-HAp paste. The study represents a step forward for preventive dentistry, concluding that pasta-PGA / nano-HAp paste as it was formed increased the microdurability of the tooth and its preventive efficiency.

The references do respect any style. BUT It's unacceptable that the entire manuscript has no citations. 

Reviewer 2 Report

This study presents a serious logical narrative based on main subject and a methodological bias. In all parts of the study, there is a mixture of ideas related to caries lesion, erosive lesion, and etched enamel. The methodology is based on etched enamel with 1M acetic acid application for 3min, which doesn’t correspond to erosive lesion widely discussed and main objective of this study. As well as the methodology wasn’t confirmed by scientific references.

This methodology doesn’t correspond a clinical situation and eliminates the protective factor of saliva.

In addition, the statistical analysis performed are inadequate due to the continuation of the analysis for 10, 20 and 30min. Besides the paste factor (nano-Hap and GC tooth mousse) was not taken into consideration, however it was extensively discussed.

Finally, a hard justification for a small sample size should be provided prior the submission.